# Progression Rate from the Indeterminate Form to the Cardiac Form in Patients with Chronic Chagas Disease: Twenty-Two-Year Follow-Up in a Brazilian Urban Cohort

**DOI:** 10.3390/tropicalmed5020076

**Published:** 2020-05-12

**Authors:** Alejandro Marcel Hasslocher-Moreno, Sergio Salles Xavier, Roberto Magalhães Saraiva, Luiz Henrique Conde Sangenis, Marcelo Teixeira de Holanda, Henrique Horta Veloso, Andrea Rodrigues da Costa, Fernanda de Souza Nogueira Sardinha Mendes, Pedro Emmanuel Alvarenga Americano do Brasil, Gilberto Marcelo Sperandio da Silva, Mauro Felippe Felix Mediano, Andrea Silvestre de Sousa

**Affiliations:** Evandro Chagas National Institute of Infectious Disease, Oswaldo Cruz Foundation, Rio de Janeiro 21040-900, Brazil; sergio.xavier@ini.fiocruz.br (S.S.X.); roberto.saraiva@ini.fiocruz.br (R.M.S.); luiz.sangenis@ini.fiocruz.br (L.H.C.S.); marcelo.holanda@ini.fiocruz.br (M.T.d.H.); henrique.veloso@ini.com.fiocruz (H.H.V.); andrea.costa@ini.fiocruz.br (A.R.d.C.); fernanda.sardinha@ini.fiocruz.br (F.d.S.N.S.M.); pedro.brasil@ini.fiocruz.br (P.E.A.A.d.B.); gilberto.silva@ini.fiocruz.br (G.M.S.d.S.); mauro.mediano@ini.fiocruz.br (M.F.F.M.); andrea.silvestre@ini.fiocruz.br (A.S.d.S.)

**Keywords:** Chagas disease, heart disease, electrocardiogram, disease progression

## Abstract

Most patients with chronic Chagas disease (CD) present the indeterminate form and are at risk to develop the cardiac form. However, the actual rate of progression to the cardiac form is still unknown. Methods: In total, 550 patients with the indeterminate CD form were followed by means of annual electrocardiogram at our outpatient clinic. The studied endpoint was progression to cardiac form defined by the appearance of electrocardiographic changes typical of CD. The progression rate was calculated as the cumulative progression rate and the incidence progression rate per 100 patient years. Results: Thirty-seven patients progressed to the CD cardiac form within a mean of 73 ± 48 months of follow-up, which resulted in a 6.9% cumulative progression rate and incidence rate of 1.48 cases/100 patient years. Patients who progressed were older (mean age 47.8 ± 12.2 years), had a higher prevalence of associated heart diseases (*p* < 0.0001), positive xenodiagnosis (*p* = 0.007), and were born in the most endemic Brazilian states (*p* = 0.018). Previous co-morbidities remained the only variable associated with CD progression after multivariate Cox proportional hazards regression analysis (*p* = 0.002). Conclusion: The progression rate to chronic CD cardiac form is low and inferior to rates previously reported in other studies.

## 1. Introduction

Chagas disease (CD) is considered a neglected tropical disease by the World Health Organization, with an estimated 8 million people infected worldwide [1]. As a result of globalization, cases are no longer restricted to Latin America and this new paradigm is a new challenge to be overcome [2]. Control programs for vectorial and transfusional transmission of CD, developed in the 1980s, significantly decreased transmission. However, surveillance challenges remain due to new outbreaks of oral transmission in endemic countries [3] and to the possibility of vertical transmission, even in nonendemic areas [4]. The urbanization of the disease expanded access to healthcare services in endemic and non-endemic countries, changing their epidemiological profile, and together with CD control programs, led to an increase in the age range of patients [5]. Integrated surveillance and healthcare actions currently target the large number of patients already infected with *Trypanosoma cruzi* [6], a significant portion of whom may develop chronic Chagas heart disease (CHD), a major determinant of morbidity and mortality [7]. Studies on the indeterminate chronic form (ICF) of CD are usually cross-sectional, most of which were performed in rural areas and were only aimed at describing its prevalence, and rarely have prospective designs. Approximately 50% of the infected people have ICF, which is characterized by low morbidity, patients with full working capacity, and an excellent medium-term prognosis [8]. Despite the importance of these studies, their data should not be extrapolated to patients who currently live in large urban centers. 

The rate of disease progression from ICF to CHD is poorly known. The few prospective studies that addressed this issue considerably differed in the study population age, number of cases, length of follow-up, geographical area, living in endemic area with or without active vectorial transmission, and migration to urban areas. More recent studies using methods similar to those of the present study were conducted in different countries (Brazil, Argentina, and Venezuela) with different geographic, climatic, and ecosystem configurations and with different vector transmission dynamics, which may explain the differences in the reported rates of disease progression [9,10,11,12,13,14,15,16,17,18,19,20,21,22].

Thus, the objective of this study was to estimate the rate of progression from the ICF to CHD in a large Brazilian urban cohort of chronic CD patients and to identify the factors that are associated with CD progression. 

## 2. Methods

This is a retrospective observational study of a historical cohort, consisting of patients diagnosed with the ICF of CD, followed at the outpatient center of the Evandro Chagas National Institute of Infectious Diseases (Instituto Nacional de Infectologia Evandro Chagas—INI) of the Oswaldo Cruz Foundation (Fundação Oswaldo Cruz—Fiocruz), from November 1986 to December 2007 and followed until December 2008. Patients who were not followed for at least one year or without a paired electrocardiogram (ECG) during the follow-up were excluded from the study. The study was approved by the INI/Fiocruz Research Ethics Committee (054/2011).

Serological diagnosis of CD was confirmed when two serological techniques were reactive: Indirect immunofluorescence (titer >1/40) and enzyme-linked immunosorbent assay (reactivity index >1.2). All patients with a confirmed diagnosis were subjected to an initial evaluation protocol, which included: Epidemiological history, directed anamnesis, physical examination focused on CD-related cardiovascular signs and symptoms, 12-lead ECG, and two-dimensional echocardiogram with Doppler (ECHO). 

Some patients were submitted to parasitological evaluation through xenodiagnosis (xeno) as recommended by Cerisola et al. [23]. ECG was performed on admission to the cohort and repeated annually in all patients. The Minnesota Code Manual of Electrocardiographic Findings [24], modified for CD, was used to standardize the ECG interpretation. The electrocardiogram changes considered compatible with Chagas disease followed the criteria recommended by the 2nd Brazilian Consensus on Chagas Disease, 2015: 2nd- and 3rd-degree right bundle-branch block, associated or not to left anterior fascicular block; frequent ventricular premature beats; polymorphous or repetitive nonsustained ventricular tachycardia; 2nd- and 3rd-degree atrioventricular block; sinus bradycardia with heart rate 50 bpm; sinus node dysfunction; 2nd- and 3rd-degree left bundle-branch block; atrial fibrillation; electrical inactive area; or primary ST-T wave changes [25]. The echocardiographic examination included parasternal and cross-sectional views and 2-, 4-, and 3- chamber apical views and variations to identify wall motion abnormalities. Left ventricular global systolic function was assessed by the Simpson method and classified as normal, mild, moderate, or severely depressed [26]. Patients’ follow-up included at least one annual medical visit and one annual ECG.

## 3. Data Analysis

Categorical variables were described as the frequency (percentage) and numerical variables as the mean and standard deviation. The Chi-squared test was used to compare categorical variables. Cumulative incidence, which is expressed as the proportion between those who were exposed at baseline and those who presented the studied end-point during the observation period, and incidence density, which is expressed as the number of events during the time of exposure of each individual, were described in the incidence analysis. Uni and multivariate Cox analyses were performed to identify CD progression predictors. Kaplan–Meier survival curves stratified according to the presence or absence of variables associated with CD progression were constructed and compared using the log-rank test. The program IBM® SPSS® Statistics 21 (New York, NY, USA) was used, setting the significance level at 5% for all tests.

## 4. Results

Of a total of 1606 patients with CD, followed at INI/Fiocruz, from November 1986 to December 2008, 701 met the inclusion criteria. Of these patients, 151 were excluded because they were not followed for at least 1 year or because they did not perform a second ECG during the follow-up period. The final studied population consisted of 550 patients (44.2 ± 11.5 years, 48.9% men) who were followed for a mean period of 65 ± 42 months.

Most patients were born in the Bahia (BA) and Minas Gerais (MG) states, accounting for 23.3% and 22.9% of the subjects, respectively. Most patients had been away from an endemic area for more than 20 years (54.4%). At baseline, 519 (94.4%) patients had a normal ECHO. Xeno tests were performed in 107 patients, of whom 34.6% were positive. Of the 550 patients followed, 99 (18%) were treated with benznidazole at baseline.

A total of 37 patients progressed to CHD according to new ECG changes, resulting in a 6.7% cumulative incidence and 1.48 by 100 patients/year incidence density. The mean age at CHD progression was 56.2 years. Patients who progressed to CHD were older, had a longer mean follow-up time, a higher prevalence of associated heart disease, were more likely born in the Bahia and Minas Gerais states, had lived more than 20 years away from endemic areas, and showed positive xeno compared to non-progressors (Table 1). 

Among the 37 progressors, 26 (70%) presented comorbidities: 16 patients had systemic arterial hypertension (SAH), including 6 with left ventricular hypertrophy (LVH) on ECHO; 12 patients had dyslipidemia; and 4 patients presented diabetes mellitus (DM). Fifteen patients presented only one comorbidity, 6 presented with 2 comorbidities, and 5 presented with 3 comorbidities. Four cardiovascular events occurred in these patients: Acute myocardial infarction (AMI), total atrioventricular block (TAVB), atrial fibrillation (AF), and heart failure (HF). One death was associated with ischemic heart disease. 

Among the new changes diagnosed on follow-up ECGs, 67.5% were intraventricular conduction disorders (IVCD), followed by primary ST-T wave changes (32.4%). and ventricular arrhythmias (29.7%) (Table 2). In total, 30% of patients had an abnormal ECHO at baseline, and LV wall motion abnormalities and LVH were the predominant findings. There were no significant differences in alterations on ECHO between progressors and non-progressors at baseline. Among the 37 progressors, 7 were subjected to xeno, including 6 positive and 1 negative, and 6 received specific treatment with benznidazole at the beginning of their follow-up.

The five variables that initially showed differences between progressors and non-progressors at baseline were tested by univariate Cox regression, remaining associated with progression to CHD, age, associated heart disease, and living more than 20 years away from endemic areas. After the multivariate Cox regression model, including these three variables, only associated heart disease remained independently associated with progression to CHD (Table 3). Figure 1 shows the Kaplan–Meier curve stratified according to the presence of associated heart disease (log-rank test *p* < 0.001).

## 5. Discussion

Our study described a lower CD progression rate than previous studies. Among the reasons that can ascertain this difference are the different ECG criteria used to define CHD, reinfection in endemic areas, mean age at baseline, and follow-up time.

The changes in ECG considered as ECG progression criteria described in most previous studies included changes, such as secondary ST-T wave changes, first-degree atrioventricular and intraventricular conduction disorders, and isolated supraventricular and ventricular extra systoles, which are no longer considered diagnostic of CHD, as described in the guidelines followed in the present study. This fact may have contributed to an overestimation of CD progression in these studies.

Reinfection could influence CD progression to CHD among patients included in field studies conducted in the 1960s and 1970s, when CD control measures were still not effective [11]. 

The mean age of patients at baseline may also influence the difference in results between studies. Patients of a younger age have more time to present CD progression during their follow-up [27,28]. The mean age of the patients followed in our study was 44.2 years, while in field studies performed in the 1960s and 1970s, the mean age was younger than 25 years, including children and adolescents, thereby accounting for a higher number of progressions. In turn, from the 1990s, studies have tended to include adults ≥40 years and elderly people, who likely had already progressed to chronic CHD, thus accounting for the lower rate of progression found in our cohort. 

The follow-up observation time is also a key variable, which accounts for differences in the rates of progression to CHD found between studies considering the natural history of CD. A retrospective cohort study of healthy blood donors showed that the rate of progression to CHD was 1.85% per year, defining that a 10-year follow-up period would be sufficient to identify the incidence of CHD [29]. The follow-up time of published studies ranges from 3 to 13 years, while the follow-up time of our study was 22 years, which is the longest follow-up time of longitudinal studies thus far. Even our reported loss to follow-up of 22% did not decrease the value of our long-term follow-up, as 78% of our patients had a follow-up of at least 10 years.

Women slightly prevailed (51.1%) in our sample, as in other studies [11,13,18,20,21]. Male sex was associated with progression to CHD in a previous field study [30], which was not confirmed in our study.

Most patients of our cohort were long-term residents of the metropolitan region of the state of Rio de Janeiro, similar to those described in other urban cohorts [19,21,22]. However, they were mostly migrants from 19 endemic Brazilian states, mainly Bahia and Minas Gerais. These states present the highest prevalence of CD and CHD [31], which may be the result of the varying pathogenic degrees of *T. cruzi* [15]. Regional differences are associated with both disease severity and the predominance of clinical forms due to factors linked to the parasitized individual (immune status, nutritional status, genetic factors, and physical effort) and to other factors related to the parasite (different strains of *T. cruzi*, parasitism intensity, and reinfections) [32,33].

Considering the role that circulating *T. cruzi* can play in the evolution of CHD, we assessed the relationship between positive xeno and ECG progression. The frequency of positive xeno in our study was similar to that of previous studies with chronic patients [34]. Nevertheless, no association was found between positive parasitemia and ECG progression, most likely due to the limited number of patients subjected to this evaluation protocol. The absence of an association between parasitemia and chronic CD progression was also described in other longitudinal studies [35].

Regarding comorbidities, the percentage of SAH, DM, and dyslipidemia in this cohort is similar to the estimated prevalence of these comorbidities in the Brazilian population, and, as described in other series of patients with CD, none of them were associated with ECG progression [36,37]. Some studies demonstrated no risk in the coexistence of SAH and CD [38]. In our cohort, 16 patients (2.9%) had associated heart disease at baseline, most often LVH, with minimal or no degree of LV systolic dysfunction. We identified five cases of ischemic heart disease with three occurrences of acute myocardial infarction (AMI). The electrocardiographic changes eventually associated with these events (electrically inactive zones or isolated primary repolarization changes) were not considered criteria for CD progression but manifestations of the associated heart disease. Although the number of patients with associated heart disease at baseline was low, those patients had an increased risk of progression to CHD diagnosed by ECG changes typical of CD, and associated heart disease was the only independent risk factor for progression to CHD.

The most frequent ECG change found in those who progressed to CHD in our study was IVCD, followed by primary ST-T wave changes, and ventricular arrhythmias. This finding corroborates several studies in which the same ECG changes were the most frequently found in CHD [39,40]. Regarding progression to CHD, other studies described primary ST-T wave changes and ICD as the most frequent findings [9,21].

Regarding ECHO, segmental changes predominated in those who had an altered ECHO at baseline, whereas diffuse changes predominated in patients who had a normal ECHO at baseline. This suggests that echocardiographic alterations indicative of an incipient process of impaired global systolic function may be the physiopathogenic basis of the electrocardiographic changes identified among progressors. In a previous study of the same cohort, the prevalence of LV aneurysm was 24% among patients with an abnormal electrocardiogram and 2% among patients with a normal ECG [41]. In addition, diastolic functioning has been described as being altered in up to 10% of patients with a normal ECG in our institution [42]. 

Despite the progression to CHD by ECG changes, none of the 37 patients who started the follow-up with ICF and progressed to CHD showed symptoms or signs compatible with CD. The clinical cardiological events (AMI, AF, TAVB, and HF) that affected progressors were apparently more related to associated heart diseases and/or to aging. The only case of HF was related to tachycardiomyopathy associated with AF in an elderly patient with severe SAH. AF and TAVB also occurred in two elderly patients (66 and 85 years, respectively). The only death was recorded in a patient who had an AMI, which corroborates the good prognosis of patients with ICF described in longitudinal studies in which mortality rates are similar to that of individuals of the same age group without CD [43]. Regarding age, ECG changes that occur in the general population in primary care are associated with age and comorbidities [44]. Therefore, we should consider that in CD cohorts with elderly patients, the onset of new cardiovascular events could be expected due to the ageing process, including degenerative changes in the conduction system, or complications from other morbidities, such as SAH and atherosclerotic disease. In fact, age was associated with CD progression after univariate analysis. However, comorbidities are also highly prevalent among elderly individuals, and after multivariate analysis, age was no longer associated with CD progression.

The increase in life expectancy observed in individuals infected with *T. cruzi* and the migration of most of the population from rural areas to large urban centers exposes these individuals to several lifestyle habits that favor the development of chronic degenerative changes, such as obesity, insulin resistance, arterial hypertension, and dyslipidemia, which, in combination with nutritional aspects and aging, significantly increase the risk of cardiovascular events and death [45,46]. Over the past few decades, studies have increasingly shown that patients with CD have comorbidities that become increasingly more frequent as this population ages [5]. The difficulty in implementing healthy habits makes them more vulnerable to the concomitant onset of other chronic diseases [47]. Another key issue is that the higher the number of comorbidities is, the greater the adverse impact on prognosis will be with the increase in morbidity and mortality. Ischemic heart disease worsens ventricular function and can exacerbate the clinical symptoms of CD, worsening the prognosis. Importantly, coronary artery disease increases the risk of sudden death, which is already high in CHD. As in the general population, SAH is the most prevalent comorbidity, and these hypertensive patients are 40% more likely to present with HF than non-hypertensive patients [36,48]. 

The findings of the present study indicate that the rate of ICF progression to CHD is low and lower than the rates described in previous studies conducted in endemic and rural areas, albeit compatible with studies conducted from the 2000s, in urban and non-endemic areas. The presence of associated heart disease, especially LVH, was independently associated with ECG progression.

However, our study is limited by the characteristic slow progression of patients with CD and ICF, which resulted in a low number of events during the follow-up. The low number of events makes the identification of other potential variables associated with ECG progression difficult. Thus, further studies with a longer follow-up period or larger number of patients are necessary to better identify the variables associated with progression.

## Figures and Tables

**Figure 1 tropicalmed-05-00076-f001:**
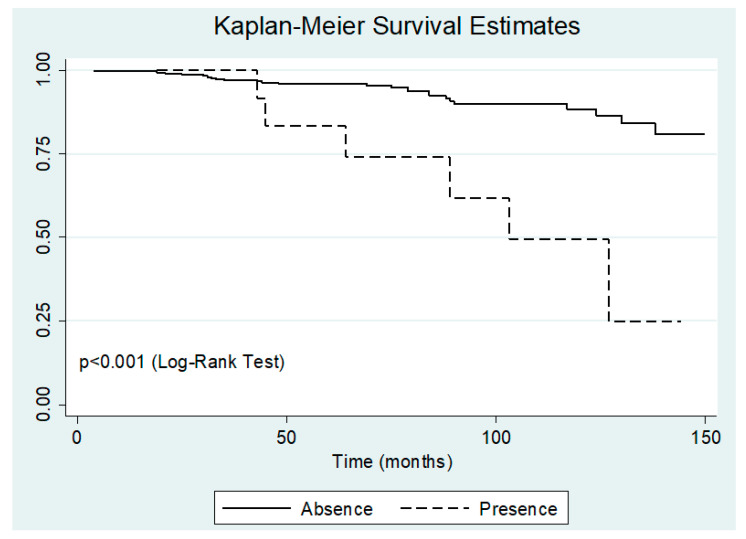
Kaplan–Meier survival curve stratified by the presence of non-chagasic cardiomyopathy.

**Table 1 tropicalmed-05-00076-t001:** Baseline characteristics of progressor and non-progressor patients. Age and follow-up time are expressed as mean ± standard deviation; ^¥^ n = 107 (7 progressors and 100 non-progressors); BA: Bahia State: MG: Minas Gerais State.

Variable	Total Patients (n = 550)	Progressors (n = 37)	Non-Progressors (n = 512)	*p*-Value
Age (years)	44.25 ± 11.55	47.8 ± 12.2	44 ± 11.5	0.05
Follow-up time (months)	65 ± 42	73 ± 48	64 ± 42	0.25
Male	269 (48.9%)	18 (48.7%)	251 (48.9%)	0.97
Hypertension	180 (32.7%)	16 (43.2%)	164 (32%)	0.16
Diabetes	23 (4.2%)	4 (10.8%)	19 (3.7%)	0.06
Dyslipidemia	133 (24.2%)	13 (35.1%)	120 (23.4%)	0.11
Associated heart disease	16 (2.9%)	6 (16.2%)	10 (1.95%)	<0.001
From State of BA/MG	254 (46.2%)	24 (64.9%)	230 (44.8%)	0.018
Living in a non-endemic area ≥20 anos	299 (54.4%)	29 (78.4%)	270 (52.6%)	0.002
Positive xenodiagnosis ^¥^	37 (34.6%)	6/7 (85.7%)	31/100 (31%)	0.007
Previous benzonidazol treatment	99 (18%)	6 (16.2%)	93 (18.1%)	0.77
Altered echocardiogram	31 (5.6%)	3 (8.1%)	28 (5.5%)	0.46

**Table 2 tropicalmed-05-00076-t002:** Age, sex, follow-up time, and electrocardiogram (ECG) and echocardiogram (ECHO) abnormalities among those patients that progressed to the cardiac form (n = 37) AF: atrial fibrillation; AFL: atrial flutter; AS: sinus arrhythmia; AVB1°: first degree atrioventricular block; EIA: electrical inactive area; FPAC = frequent premature atrial complex; FVPB: frequent ventricular premature beats; IPAC-isolated premature atrial complex; IVPB: isolated ventricular premature beats; LAFB: left anterior fascicular block; LBBB1°: first-degree left bundle-branch block; LBBB2°: second-degree left bundle-branch block; LBBB3°: third degree left bundle-branch block; LV: QRS low voltage; MAP: migratory atrial pacemaker; PRA: primary ST-T wave changes; RBBB1°: first-degree right bundle-branch block; RBBB3°: third-degree right bundle-branch block; SBRAD: sinus bradycardia with heart rate 50 bpm; SRA: secondary ST-T wave changes; ANEU: aneurism; DIS; dysfunction; LVEF: left ventricular ejection fraction; LVH: left ventricular hypertrophy; SEG DIS: segmental dysfunction.

ID	Sex	Time to Progression (Months)	Age at Beginning (Years)	Age at Progression (Years)	Initial ECG	ECG at Progression	Initial ECHO	Initial ECHO Abnormality	Initial LVEF	ECHO at Progression
1	M	48	67	69	NORMAL	PRA	NORMAL	‒	77%	NORMAL
2	M	127	45	55	NORMAL	RBBB3° + LAFB + IVPB	ABNORMAL	LVH	56%	ANEU + LVEF = 52%
3	M	75	33	39	NORMAL	PRA + RBBB2°	NORMAL	‒	63%	NORMAL
4	M	84	35	42	NORMAL	RBBB3° + FVPB	NORMAL	‒	71%	LVH
5	M	22	56	58	MAP	FVPB	ABNORMAL	SEG DIS	50%	NORMAL
6	M	43	43	45	NORMAL	RBBB2°	NORMAL	‒	67%	NORMAL
7	M	35	47	50	SBRAD + AVB1°	PRA + FVPB	ABNORMAL	SEG DIS	55%	LVH + DIS/LVEF = 40%
8	F	90	74	81	AVB1°	LAFB + AFL	ABNORMAL	SEG DIS	54%	NORMAL
9	M	89	59	66	NORMAL	AF	NORMAL	‒	64%	LVH + DIS/LVEF = 53%
10	M	178	44	59	NORMAL	PRA + AVB1° + LAFB + LBBB2°	NORMAL	‒	64%	DIS/LVEF = 54%
11	M	117	20	29	AS	LBBB3°	ABNORMAL	LVH	56%	LVH + DIS/LVEF = 49%
12	F	25	37	39	SRA	EIA	ABNORMAL	ANEU	61%	NORMAL
13	F	43	57	60	SBRAD + SRA	PRA	ABNORMAL	LVH	74%	NORMAL
14	M	31	29	31	LAFB	PRA + LAFB	NORMAL	‒	70%	NORMAL
15	M	79	20	26	NORMAL	SBRAD + AVB1° + RBBB3° + LAFB	NORMAL	‒	69%	NORMAL
16	F	103	56	63	IVPB	PRA + IPAC	NORMAL	‒	84%	NORMAL
17	F	84	42	49	IVPB	RBBB3°	ABNORMAL	LVH	73%	NORMAL
18	F	88	54	61	AVB1°	LBBB2°	NORMAL	‒	79%	LVH
19	F	44	47	51	RBBB1°	PRA +LBBB1° + LV	NORMAL	‒	72%	NORMAL
20	M	138	55	66	NORMAL	AF	NORMAL	‒	63%	NORMAL
21	F	89	60	70	LBBB1°	RBBB2° + LAFB + IVPB	ABNORMAL	LVH	75%	ANEU
22	M	33	60	63	NORMAL	EIA	NORMAL	‒	69%	NORMAL
23	M	168	50	54	NORMAL	RBBB2° + LAFB	NORMAL	‒	73%	NORMAL
24	F	124	40	50	NORMAL	PRA + FVPB	NORMAL	‒	68%	LVH + DIS/LVEF = 54%
25	F	19	48	50	SBRAD + IPAC	SBRAD + RBBB2°	NORMAL	‒	66%	NORMAL
26	F	21	43	45	LAFB + LV	RBBB2° + LAFB + LV	NORMAL	‒	74%	NORMAL
27	F	19	43	45	NORMAL	RBBB2° + LAFB	NORMAL	‒	64%	LVH
28	F	192	50	66	NORMAL	PRA + SBRAD	NORMAL	‒	68%	NORMAL
29	F	30	31	34	SBRAD + LBBB1°	PRA + LAFB + FVPB	NORMAL	‒	65%	NORMAL
30	F	4	64	64	SRA	LBBB3°	NORMAL	‒	60%	NORMAL
31	M	45	57	61	AS	MAP +LBBB1° + IPAC + FVPB	ABNORMAL	LVH	74%	DIS/LVEF = 52%
32	M	32	37	39	SBRAD	LBBB3° + MAP	NORMAL	‒	63%	NORMAL
33	M	64	60	65	SRA	RBBB3° + LAFB + FVPB	ABNORMAL	LVH	58%	NORMAL
34	F	31	50	52	NORMAL	PRA	ABNORMAL	LVH	74%	NORMAL
35	F	79	54	60	NORMAL	RBBB2°	NORMAL	‒	62%	NORMAL
36	F	69	52	57	NORMAL	RBBB2° + FVPB	NORMAL	‒	65%	NORMAL
37	F	130	50	55	NORMAL	RBBB3° + LAFB + FVPB	NORMAL	‒	64%	NORMAL

**Table 3 tropicalmed-05-00076-t003:** Univariate and multivariate Cox regression model for progression from the indeterminate to cardiac form.

Variable	Univariate Analysis		Multivariate Analysis	
	HR (95% CI)	*p*-Value	HR (95% CI)	*p*-Value
Age	1.03 (1.00 to 1.06)	0.03	1.02 (0.98 to 1.05)	0.37
Associated heart disease	5.37 (2.22 to 13.00)	<0.001	4.10 (1.65 to 10.20)	0.002
From State of BA/MG	1.68 (0.85 to 3.34)	0.14	−	−
Living in a non-endemic area ≥20 years	2.44 (1.10 to 5.38)	0.03	1.81 (0.77 to 4.26)	0.18
Positive xenodiagnosis	1.07 (0.35 to 3.29)	0.89	−	−

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
