# Peer review of "Progression Rate from the Indeterminate Form to the Cardiac Form in Patients with Chronic Chagas Disease: Twenty-Two-Year Follow-Up in a Brazilian Urban Cohort"

_tropicalmed, 2020, doi:10.3390/tropicalmed5020076_

Round 1
Reviewer 1 Report
dear author
This is a nicely presented longitudinal cohort I really read with attention and very easily.
I think it can be accepted in the present form.
Author Response
We are grateful to the reviewer’s and editor’s comments.
Reviewer 2 Report
The authors have conducted a well-designed study to determine the risks of developing and progression of electrocardiographic abnormalities associated with chronic T. cruzi infection in a cohort of patients who migrated from rural to urban areas of Brazil.
Author Response

(The authors gave the same response as above.)

Reviewer 3 Report
- Of the 26 patients that progressed to CHD and had comorbidities, how many had multiple comorbidities?
- In Table 2
- Please fix the date of start and/or date of progression in cohort for sample IDs 2, 17, 23, 32.
- Please define echo acronyms ANEU, DIS, DIS SEG, HIP DIF
Author Response
1. Of the 26 patients that progressed to CHD and had comorbidities, how many had multiple comorbidities?
The requested information was included in the revised text.
2. In Table 2
- Please fix the date of start and/or date of progression in cohort for sample IDs 2, 17, 23, 32.
We did the correction. Please, see the revised Table 2.
- Please define echo acronyms ANEU, DIS, DIS SEG, HIP DIF
We did the correction. Please, see the revised Table 2.
